

# Identify devices and events from non-IP heterogeneous IoT network traffic

Yi Chen[1,2], Junxu Lai[1], Zhu Lin[1], Meijing Zhang[1,2] and Wenxi Liu[2,3]

[1] Fujian Police College, Department of Computer and Information Security Management, Fu Zhou, Fu Jian, China
[2] Fujian Police College, Collaborative Innovation Research Center of Intelligent Policing, Fu Zhou, Fu Jian, China
[3] Fuzhou University, College of Computer and Data Science, Fu Zhou, Fu Jian, China

## ABSTRACT

In recent years, notable advancements have been achieved in the realm of identifying IP-based Internet of Things (IoT) devices and events. Nevertheless, the majority of methods rely on extracting fingerprints or features from plain text IP-based packets, which limits their ability to accommodate heterogeneous IoT devices such as ZigBee and Z-Wave, and fails to address the challenge of limited traffic samples. To tackle these issues, we propose a novel approach based on IoT communication characteristics and featuring module extensibility. This method is presented to effectively identify IoT devices and events from non-IP heterogeneous IoT network traffic. To shield the differences caused by the heterogeneous IoT protocol, a heterogeneous sample extraction platform with an extensible structure is created to extract raw sequence samples from ZigBee and Z-Wave traffic, with potential for expansion to other protocols. To address the challenges arising from the scarcity of samples, a sample identification framework based on IoT communication characteristics is devised to create synthetic samples from the raw sequence samples, enabling concurrent processing of the raw and synthetic samples using an identification model featuring two separate sequence networks. Comparative assessments of our method against baseline sequence models and the latest techniques demonstrate the advantages of our approach in identifying non-IP heterogeneous IoT traffic. The experimental results indicate that our method achieves an average accuracy improvement of 29.7% compared to baseline models using only raw samples. Furthermore, our method shows improvements of 22.1%, 21.5%, and 21.8% in macro precision, macro recall, and macro F1-score, respectively, over the latest method.

# INTRODUCTION

With the advancement of IoT technology, an increasing number of commercial IoT products are deployed in various scenarios, such as homes and office spaces. Due to the diverse nature of IoT products, the lack of a unified security mechanism, and their close association with human activities, IoT devices gradually become prime targets for attackers (*Hassija et al., 2019*). To enhance the security of existing IoT networks, numerous researchers explore methods for identifying IoT devices and events from IoT network traffic (*Sánchez et al., 2021*).

Corresponding author
Meijing Zhang, zmjm2@163.com

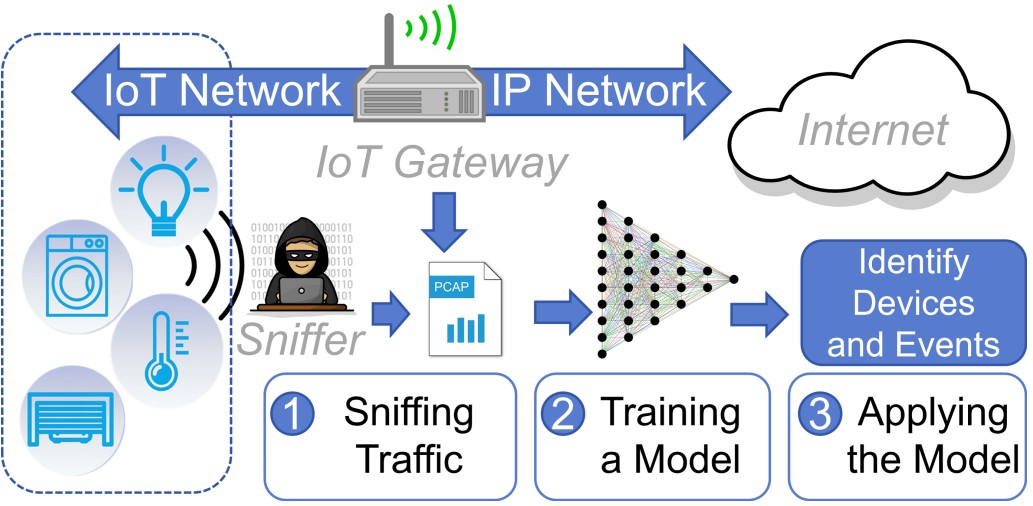

**Figure 1** The application scenario of a device identification method.

As illustrated in Fig. 1, assuming that a heterogeneous IoT network is connected to the Internet through a gateway, an identification method can sniff traffic from the gateway or the wireless network and model the traffic. Subsequently, the model can be employed to identify device types or operational events from the traffic. In the realm of network security, device identification technology can be employed to discover potential IoT devices in the environment, preventing low-security devices from serving as gateways for attackers to intrude the intranet (*Miettinen et al., 2017*; *Kostas, Just & Lones, 2022*; *Kostas, Just & Lones, 2023*; *Marchal et al., 2019*). Regarding privacy leakage, these methods can be applied to detect state change events of IoT devices, potentially infringing upon users' private lives (*Copos et al., 2016*; *Shafqat et al., 2022*; *Acar et al., 2020*; *Apthorpe et al., 2019*). For instance, it is possible to determine whether a user is at home based on the lights and even infer the user's personality based on their device usage habits (*Copos et al., 2016*). Therefore, the identification of IoT devices and events remains a significant research focus (*Jmila et al., 2022*; *Chowdhury & Abas, 2022*).

Recently, significant progress has been made in the identification of IP-based IoT devices and events. For example, *Miettinen et al. (2017)* propose an identification system, IOT SENTINEL, based on device fingerprint matching and edit distance. *Kostas, Just & Lones (2022)* devise a machine learning (ML)-based method, IoTDevID, which introduces a set of traffic feature extraction methods and corresponding packet aggregation algorithms to collaborate with ML models for packet identification, improving the performance of the detection model. *Qu et al. (2023)* propose an input-agnostic hierarchical deep learning framework, leveraging deep learning techniques to learn device fingerprint features directly from encrypted packets. However, the methods in *Miettinen et al. (2017)* and *Kostas, Just & Lones (2022)* both require extracting fingerprints or features from plaintext IP-based packets, rendering them unable to support heterogeneous IoT devices like ZigBee and Z-Wave, and incapable of directly processing encrypted packets. Although the method

in *Qu et al. (2023)* demonstrates the capability to handle heterogeneous network traffic, the deep learning methods utilized in this approach encounter significant challenges of overfitting in scenarios with limited samples, since low-power IoT devices typically generate minimal communication packets to prolong battery life.

In order to handle the aforementioned limitations, we propose a novel approach featuring module extensibility and based on IoT communication characteristics. It primarily consists of the heterogeneous sample extraction platform and the sample identification framework. The advantages lie in: Firstly, this platform can shield the differences caused by the heterogeneous IoT protocol stack. It can parse heterogeneous IoT packets like ZigBee, Z-Wave, *etc.*, and generate serialized numerical samples that can be processed by the framework. Additionally, it can be expanded to support new protocols. Second, the sample identification framework is optimized based on the communication characteristics of IoT traffic, making it more suitable for scenarios with very few samples in heterogeneous IoT datasets. Third, this method does not rely on the plaintext content of packets, allowing direct processing of encrypted traffic from various heterogeneous IoT protocols.

Specifically, firstly the heterogeneous sample extraction platform reads the traffic content and transforms each packet into a 4-tuple consisting of relative time, sender direction, payload length, and message type, which are present in any heterogeneous IoT traffic. Since all 4-tuples of packets are transformed into corresponding integers by our proposed algorithm, all traffic samples are processed to create a raw sample dataset composed of integer sequences. Then, our designed packet synthesis algorithm is utilized to re-encode the repeated identical subsequences in the raw samples to generate new synthetic samples. This algorithm effectively reduces the impact of the Stop-and-Wait protocol of IoT devices on identification. Afterwards, the raw samples and the synthetic samples will be input into the model simultaneously. These samples are individually processed through dedicated embedding layers for encoding and then passed into sequence models (such as BiLSTM, BiGRU) for computation to produce the respective vector representations. Subsequently, the two vectors are concatenated and fed into a multilayer perceptron (MLP) network for multi-classification.

In order to evaluate the effectiveness of our method, we conduct two sets of experiments. In the first set, we use three independent datasets (ZigBee from CICIOT2022 (*Dadkhah et al., 2022*), Z-Wave from CICIOT2022 (*Dadkhah et al., 2022*), and ZigBee from ZLeak (*Shafqat et al., 2022*)) and implement sample identification models using five sequence networks including BiGRU, BiLSTM, BiLSTM-ATT, BiRNN, and CONV-1D. We evaluate the performance of these models by classifying raw samples, synthetic samples, and a combination of raw and synthetic samples. The results from these experiments demonstrate that our method significantly enhances the detection performance of existing baseline sequence models. Additionally, it validates the effectiveness of the heterogeneous sample extraction platform. In the second set, we conduct a comparative evaluation between our method, IoTDevID (*Kostas, Just & Lones, 2022*), and trace-classifier (trace-clf) (*Qu et al., 2023*; *Jiang, 2024*) on the ZigBee dataset from CICIOT2022. Our method exhibits superior performance in identifying heterogeneous IoT traffic samples compared to these similar methods.

In summary, the contributions of this article are as follows:

- We develop a platform for extracting samples from heterogeneous IoT network traffic. It can extract raw sequence samples from ZigBee traffic transmitted in ZigBee Encapsulation Protocol (ZEP) tunnels, ZigBee traffic carried in IEEE 802.15.4 frames, or Z-Wave traffic in ZLF format files. Furthermore, it allows for expansion to new heterogeneous IoT protocols.

- We propose a sample identification framework based on the communication characteristics of IoT traffic. The framework involves generating synthetic samples from the raw sequence samples using a packet synthesis algorithm and then processing the raw and synthetic samples simultaneously with the identification model using two independent sequence networks, effectively improving the detection performance of the model.

- Our experiments are conducted on three independent heterogeneous IoT traffic datasets to validate that our method enhances the detection performance of existing sequence models on heterogeneous IoT traffic samples. Subsequently, a comparative evaluation of our method with the latest methods is conducted to validate the advantages of our method over similar methods in heterogeneous IoT traffic sample identification.

## RELATED WORKS

In this section, we investigate the latest advances in device identification technology and analyze the unique challenges faced in performing device identification in heterogeneous IoT network traffic.

At present, there are a large number of IoT devices in the market using WiFi standards and TCP/IP for communication. Many studies focus on the identification of these TCP/IP devices. *Kotak & Elovici (2021)* study how to use small images built from the IoT device network traffic payloads to represent the communication behavior of IoT devices, and train a MLP classifier to identify different IoT devices. *Zahid et al. (2022)* propose a framework-based hierarchical deep neural network to distinguish IoT devices from non-IoT devices using a feature set of TCP/IP headers and traffic. *Aksoy & Gunes (2019)* propose a method that combines sensor measurements and statistical feature sets from TCP/IP headers to extract features for identifying IoT devices. *Pinheiro et al. (2019)* select the statistical mean, standard deviation, and number of bytes transmitted within a one-second window of IP packets as sample features, and utilize ML methods such as random forest for classification. *Ortiz, Crawford & Le (2019)* utilize stacked autoencoders to automatically learn features from device traffic, learn the classes of traffic observed, and classify IoT device traffic accordingly. In addition to directly utilizing TCP/IP protocol fields, some studies also extract features from upper-layer protocols that rely on TCP/IP. *Perdisci et al. (2020)* study how to treat the resolved domain names as words and consider the set of domain names queried by devices as a document. Then, natural language processing algorithms are utilized to identify IoT devices from DNS traffic. *Le et al. (2019)* develop a ML-based identification method that builds a fingerprint database using device DNS traffic, and then identifies devices through the TF-IDF algorithm. *Chowdhury et al. (2020)* utilize TCP/IP

packet header features to create device fingerprints and propose a set of three metrics to separate certain features from packets that actively contribute to device identification. These features are then input into ML algorithms for classification. Other studies (*Salman et al., 2022*; *Wang et al., 2022*; *Charyyev & Gunes, 2020b*; *Thom et al., 2022*; *Luo et al., 2022*; *Charyyev & Gunes, 2020a*) also commonly involve selecting features from the TCP/IP headers and using ML algorithms to classify device types. Although these methods achieve quite ideal results, they also exhibit apparent limitations in terms of non-IP scenarios.

Due to the low-power and mesh networking requirements of IoT devices, there are a large number of IoT devices in the market based on heterogeneous network protocols such as ZigBee and Z-Wave. For ZigBee devices, their application data is generally encapsulated in the ZigBee network layer format and directly transmitted in the air using the IEEE 802.15.4 protocol or transmitted after encapsulation through the ZEP *via* UDP. For Z-Wave devices, messages use a proprietary protocol stack not publicly disclosed by Sigma Designs, requiring dedicated capture and analysis software. In addition, there are also commercial products such as the nRF24 series chips from Nordic Semiconductor (*Gheorghiu et al., 2023*). Therefore, the identification methods for Wi-Fi-based IoT devices are often difficult to apply to heterogeneous IoT networks for device identification.

In the method of identifying non-IP heterogeneous IoT devices based on packet parsing, *Shafqat et al. (2022)* study how to conduct packet unpacking analysis on ZigBee packets captured from the wireless environment and parse packets field by field to analyze the operational status of devices. *Gvozdenovic et al. (2024)* propose a method for enumerating IoT devices in the network, which includes a series of passive, active, multichannel, and multiprotocol scanning algorithms to discover IoT devices. *Nkuba et al. (2023)* parse Z-Wave packets to determine whether devices are susceptible to wireless injection attacks. These methods can support non-IP heterogeneous IoT devices such as ZigBee and Z-Wave with high accuracy and low overhead. However, the method of packet parsing is often tightly coupled with device protocols. Considering the diversity of heterogeneous IoT network protocols, as well as protocol version differences and implementation disparities among different manufacturers, maintaining a packet analysis tool that supports all protocols, versions, and products from different manufacturers is challenging. Moreover, protocol analysis must be conducted with communication secret keys, which can be difficult to achieve in certain scenarios, such as instances involving privacy leaks.

In the method of identifying non-IP heterogeneous IoT devices based on machine learning, the IoTDevID (*Kostas, Just & Lones, 2022*) considers NonIP devices. It parses 111 features from the packets and designs a unique method to select the most critical features. It then utilizes a packet aggregation algorithm in combination with ML models for device identification. IoTDevID is capable of identifying ZigBee packets transmitted after encapsulation through the ZEP *via* the UDP (*Kostas, Just & Lones, 2022*). Furthermore, an input-agnostic hierarchical deep learning frame is designed (*Qu et al., 2023*). Its advantage lies in the ability to adjust the shape of input samples based on the actual shape collected from the network, thus allowing adaptation to different sample features in heterogeneous IoT networks for device identification. *Cheng et al. (2022)* propose a device fingerprint identification scheme based on deep learning for IoT devices with

Z-Wave protocol, and propose the concept of confidence interval to solve the problem of overlapping identification of similar devices. However, due to the low-power characteristics of heterogeneous IoT devices, these devices often generate only a small number of communication samples, posing a challenge for the above-mentioned ML-based methods due to insufficient training sample quantities.

In conclusion, the challenges faced in the issue of device identification in heterogeneous IoT networks are as follows: (1) Due to the significant differences between protocols in heterogeneous IoT networks, identification methods often struggle to be applied to various heterogeneous IoT devices. (2) These ML-based methods encounter significant challenges of overfitting in scenarios with limited samples of low-power heterogeneous IoT devices.

To address the challenge posed by the differences in heterogeneous IoT protocol stacks, our method involves the design of a heterogeneous sample extraction platform to create a uniform sample structure for different protocols. This platform is used to generate samples from heterogeneous IoT traffic and transforms each packet into a 4-tuple consisting of relative time, sender direction, payload length, and message type, all of which are present in any heterogeneous IoT traffic. Additionally, to tackle the issue of limited samples, we devise a sample identification framework based on IoT communication characteristics to reduce the impact of retransmitted packets on the samples, thereby enhancing the detection performance of heterogeneous IoT traffic samples.

## PROPOSED METHOD

In this work, our goal is to propose a method for identifying specific IoT devices and their corresponding operational events from encrypted traffic in non-IP heterogeneous IoT networks, sniffed either from the gateway or wireless network. Our method primarily consists of the heterogeneous sample extraction platform and the sample identification framework. To mitigate the impact of heterogeneous IoT protocols, we develop the platform with an extensible structure to extract raw sequence samples from ZigBee and Z-Wave traffic, and has the potential for expansion to other protocols. Additionally, to address the challenges arising from the scarcity of samples, we propose the framework based on IoT communication characteristics. This framework creates synthetic samples from the raw sequence samples, enabling concurrent processing of the raw and synthetic samples using an identification model featuring two separate sequence networks.

### Heterogeneous sample extraction platform

The heterogeneous sample extraction platform is developed to extract samples from IoT network traffic. As illustrated in Fig. 2, initially, the captured network traffic sample files (*e.g.*, pcapng, zlf format files) are stored in their respective directories, with each directory containing samples from the same type of device or operational event. After traversing through the directory files, the platform processes all traffic sample files into serialized numerical samples, with the directory names serving as labels. During the processing of a sample file, the platform will invoke the corresponding heterogeneous IoT traffic processing module for unpacking, where the module reads the file content and transforms each packet into a 4-tuple consisting of relative time, sender direction, payload length, and message

  

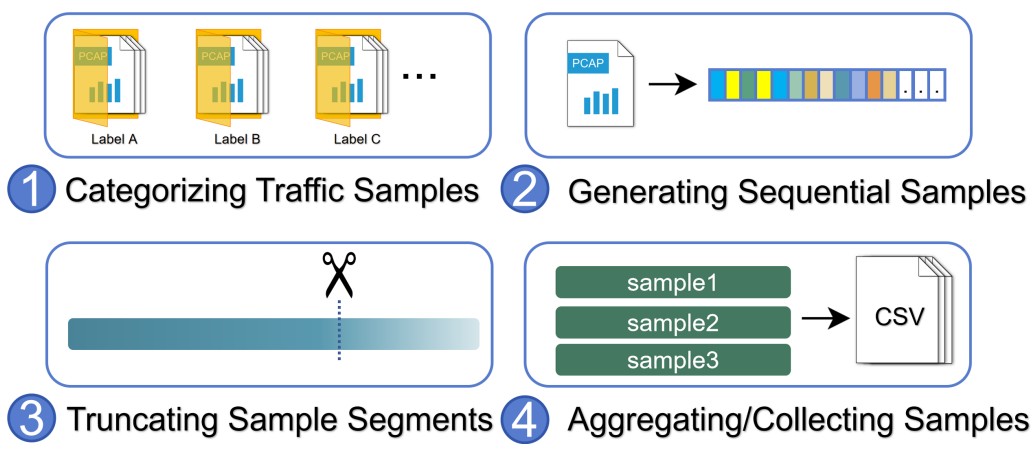

**Figure 2** The workflow of the heterogeneous sample extraction platform.

type. Eventually, each file is transformed into an ordered sequence composed of 4-tuples. Considering the need to capture a fixed duration of traffic as test samples during the model working in a deployment environment, we introduce the hyperparameter $st$ that limits each sample segment to contain at most $st$ seconds of packets. Thus, the 4-tuple sequence is divided into independent sub-sequences with $st$-second intervals. After processing all network traffic sample files, all sample sequences are saved to a dataset file for subsequent model processing.

Normally, the traffic sample files used in this workflow need to be collected manually. For example, to capture the devices' power-on traffic, all devices are unplugged and the network is rebooted. Then, each device is powered on individually. After a fixed waiting period, the network traffic is captured in isolation. This process is repeated multiple times. Similarly, to collect traffic for specific operational events, interactions with each device are performed, and the corresponding traffic is captured. In this work, we use raw traffic sample files from public datasets. These files are labeled with the device and operation that generated the traffic. Additionally, how different traffic samples are labeled depends on the actual objectives. For example, if the model is used to identify devices from traffic, the power-on, power-off, and other operational events of the same device share the same label, which represents a single device. If the model is used to identify operational events, then different labels are assigned to various operational events, with each label representing a type of operation.

The key to masking the differences in heterogeneous IoT protocol stacks lies in using different traffic processing modules to handle corresponding types of traffic sample files. These modules need to implement an interface that reads the input file and returns a 4-tuple sequence of the corresponding packets' information. The elements in the tuple contain information present in any heterogeneous IoT traffic. In IV, we implement three processing modules for handling different heterogeneous IoT traffic from three independent datasets, demonstrating the platform's usability.

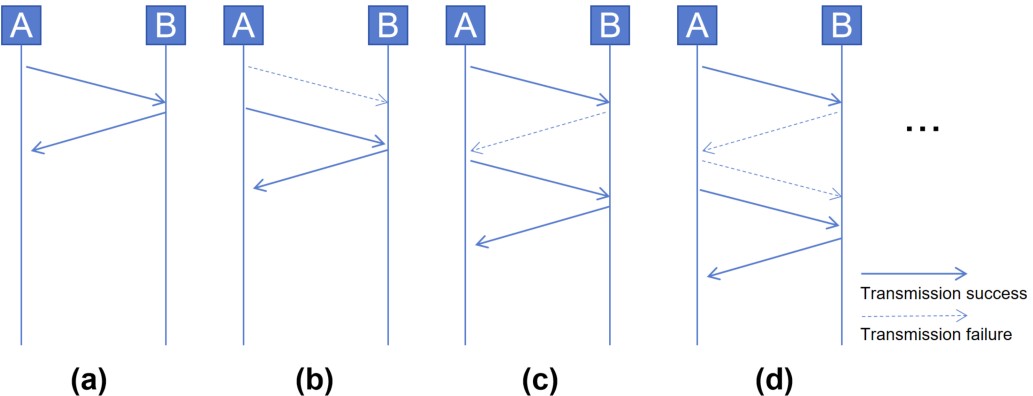

**Figure 3** Cases in Stop-and-Wait protocol: (A) normal acknowledgment, (B) retransmission on no acknowledgment, (C) and (D) complex retransmission with lost acknowledgments.

## Sample Identification framework
### Communication characteristics of IoT traffic

Stop-and-Wait protocol: In low-speed wireless IoT networks, due to the widespread use of the 2.4 GHz public frequency band and the low wireless transmission power, it is common for transmitted packets to be easily lost. Therefore, non-IP heterogeneous IoT devices often utilize a Stop-and-Wait protocol to guarantee packet delivery to the receiver. In normal cases, sender A sets a retransmission timer after sending a packet. Upon receiving the packet, receiver B sends an acknowledgment packet to inform sender A of its reception. Then A cancels its retransmission timer, as shown in Fig. 3A. In a common retransmission process, if the packet sent by A is lost, B will not send an acknowledgment packet. After the retransmission timer expires, A will retransmit the packet, as shown in Fig. 3B. However, this process may become more intricate due to the loss of acknowledgment packets. As shown in Figs. 3C and 3D, a higher number of retransmissions is required to ensure packet delivery.

The Stop-and-Wait protocol of non-IP heterogeneous IoT devices results in variations among similar network traffic samples. Similarly, it also causes duplicate packets to be retransmitted within the same sample.

Sleep strategy: Battery-powered IoT devices typically strive to remain in a sleep state to minimize battery consumption. IoT devices are usually in a sleep state, driven to work by preset timers and external interrupts. Timers are used to periodically wake up the CPU to check sensor status, or to periodically send heartbeat messages to report its own status. External interrupts often originate from sensor peripherals, which actively wake up the CPU to promptly handle unexpected new events. As a result, messages sent by IoT devices are not continuous, but often have periodic time intervals. For instance, a device may enter sleep mode after completing message transmission within 1–2 s, and then be woken up again after a longer interval to resume frequent message exchange.

### Synthesis algorithm

The synthesis algorithm uses the 4-tuple sequence $S$ outputted by the heterogeneous sample extraction platform as the input to generate an integer sequence $I$ along with their corresponding time intervals $T$. $i \in I$ represents the information extracted from a packet in $S$, while $interval \in T$ indicates the time interval between two adjacent packets in $S$. The process of generating integer sequences is described in Procedure 1. In this procedure, $pkt$ is used to iterate through each 4-tuple element of $S$, which represents the information of each IoT network packet. The sign of the integer $i$ depends on the sending direction of the packet represented by $pkt$, and the magnitude of $i$ is equal to the length of the packet. Since there are significant differences in meaning between broadcast packets, link control packets, and regular data packets, this procedure appends a larger offset value to such special packets to differentiate them from regular packets. Finally, the generated integer $i$ is appended to the end of sequence $I$, and its time $interval$ with the preceding element is appended to the end of sequence $T$.

---

**Procedure 1:** Generate Integer Sequences

**input** : a 4-tuple sequence $S$
**output:** an integer sequence $I$ along with their corresponding time intervals $T$

1   $I \leftarrow \emptyset$;
2   $T \leftarrow \emptyset$;
3   $t_p \leftarrow 0$;
4   **foreach** *element* $pkt = \langle d, l, p, t \rangle$ *of the* $S$ **do**
     `// d,l,p,t represent the sender direction, length, type, and`
     `timestamp of the pkt.`
5      **if** $d = SEND$ **then** $i \leftarrow 1 \times l$;
6      **else** $i \leftarrow -1 \times l$;
7      **if** $p = BROADCAST$ **then** $i \leftarrow i + 10^4$;
8      **if** $p = LINKCONTROL$ **then** $i \leftarrow i + 10^5$;
9      $I \leftarrow I \cup \{i\}$;
10     $T \leftarrow T \cup \{t - t_p\}$;
11     $t_p \leftarrow t$;

---

    The synthesis algorithm uses the integer sequence $I$ as the raw sample to generate a synthetic sample $Y$. The process of generating synthetic samples is described in Procedure 2. During initialization, the algorithm establishes an empty dictionary $Dict$ to save $index$: $subsequence$ pairs, which is global for all raw samples. First, based on the characteristics of IoT sleep strategy, the packets with a time $interval$ greater than $N$ seconds are split into subsequences divided by the time of $N$ seconds. As analyzed in the Sleep Strategy, a device may enter sleep mode within 1–2 s after completing message transmission, and then be woken up again after a longer period. In this algorithm, $N$ represents the threshold for the time interval from when the device starts entering wake mode until it transitions into sleep

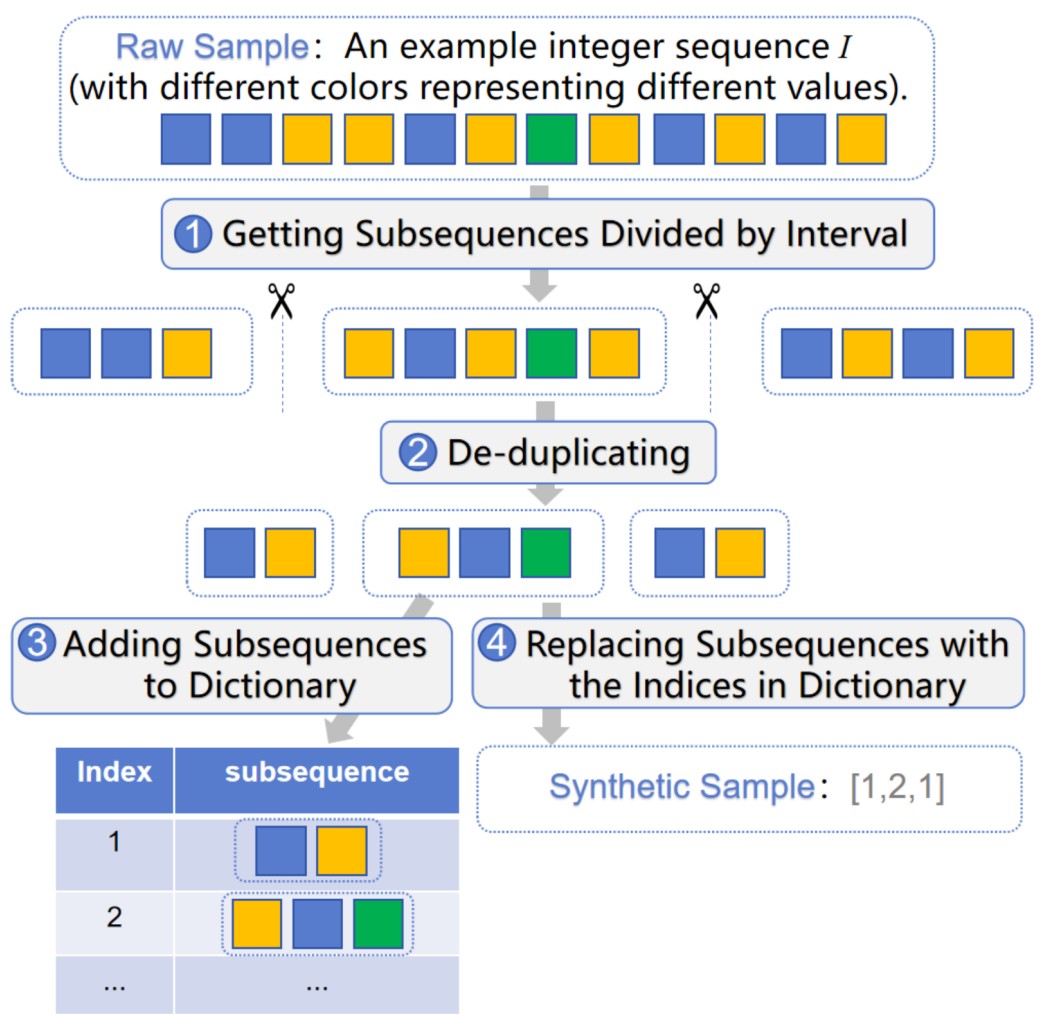

**Figure 4** **The workflow of the synthesis algorithm for generating synthetic samples from raw samples.**

mode, which can be used to divide the related packets into the same *subsequence* according to the wake period. Afterwards, the algorithm removes duplicate elements within the *subsequence* and adds the *subsequence* to *Dict* if it is not already present. Subsequently, the algorithm replaces all subsequences in the raw sample *I* with their corresponding indices from the dictionary *Dict* to generate the final synthetic sample *Y*. The workflow of the algorithm is depicted in Fig. 4.

### Identification model

The reason for deduplicating within the subsequence is that the IoT device Stop-and-Wait protocol may result in duplicate elements within the subsequence. As a result, even logically identical operations may exhibit different traffic due to fluctuations in network quality. Thus, deduplicating within the subsequence alleviates the impact of retransmitted packets on the samples.

---

**Procedure 2:** Generate Synthetic Samples

    **input** : a raw sample $I$ along with their corresponding time intervals $T$
    **output:** a synthetic sample $Y$

1   $Dict \leftarrow \{\}$;
    `// Dict is a dictionary`
2   $Y \leftarrow \emptyset$;
3   $S \leftarrow \emptyset$;
4   **for** $i \leftarrow 0$ **to** $|I| - 1$ **do**
5      $i \leftarrow I_i$;
       `// `$T_i$` represents the time difference between packets.`
6      **if** $T_i < N$ **then**
7         $S \leftarrow S \cup \{i\}$;
8      **else**
9         $S \leftarrow S \cup \{i\}$;
10        $S \leftarrow \text{Deduplicate}(S)$;
11        **if** $\langle S \rangle \notin Dict$ **then**
12           $\text{DictAddEntry}(Dict, \langle S \rangle)$;
13        $index \leftarrow \text{DictFindIndex}(Dict, \langle S \rangle)$;
14        $Y \leftarrow Y \cup index$;
15        $S \leftarrow \emptyset$;

---

The synthesis algorithm is inspired by the idea of convolutional neural networks. Convolutional networks extract fundamental features through convolutional layers and use pooling layers to enable the model to observe samples from a broader perspective. However, due to the limited number of samples in non-IP heterogeneous IoT networks, it is challenging for the convolutional network to directly learn their communication characteristics. Therefore, our method directly uses subsequences divided by time interval to extract the fundamental features of the samples, which often appear multiple times in different samples. Then, by encoding the subsequences into indexes, the model observes the overall picture of the samples from a wider range, thereby improving the model's identification ability.

As illustrated in Fig. 5, in the structure of the identification model, the raw samples output by the heterogeneous sample extraction platform and the synthetic samples generated by the synthesis algorithm are simultaneously input into the model. These samples are individually processed through dedicated embedding layers for encoding and then passed into sequence models (such as BiLSTM, BiGRU) for computation to produce their respective vector representations. Subsequently, the two vectors are concatenated and fed into a MLP network for multi-classification. Given that the number of labels is usually greater than 2 in this context, the SoftMax function is employed to generate probability distributions, thereby enhancing the model's classification performance. All the trainable parameters in the model are jointly trained for the multi-classification task.

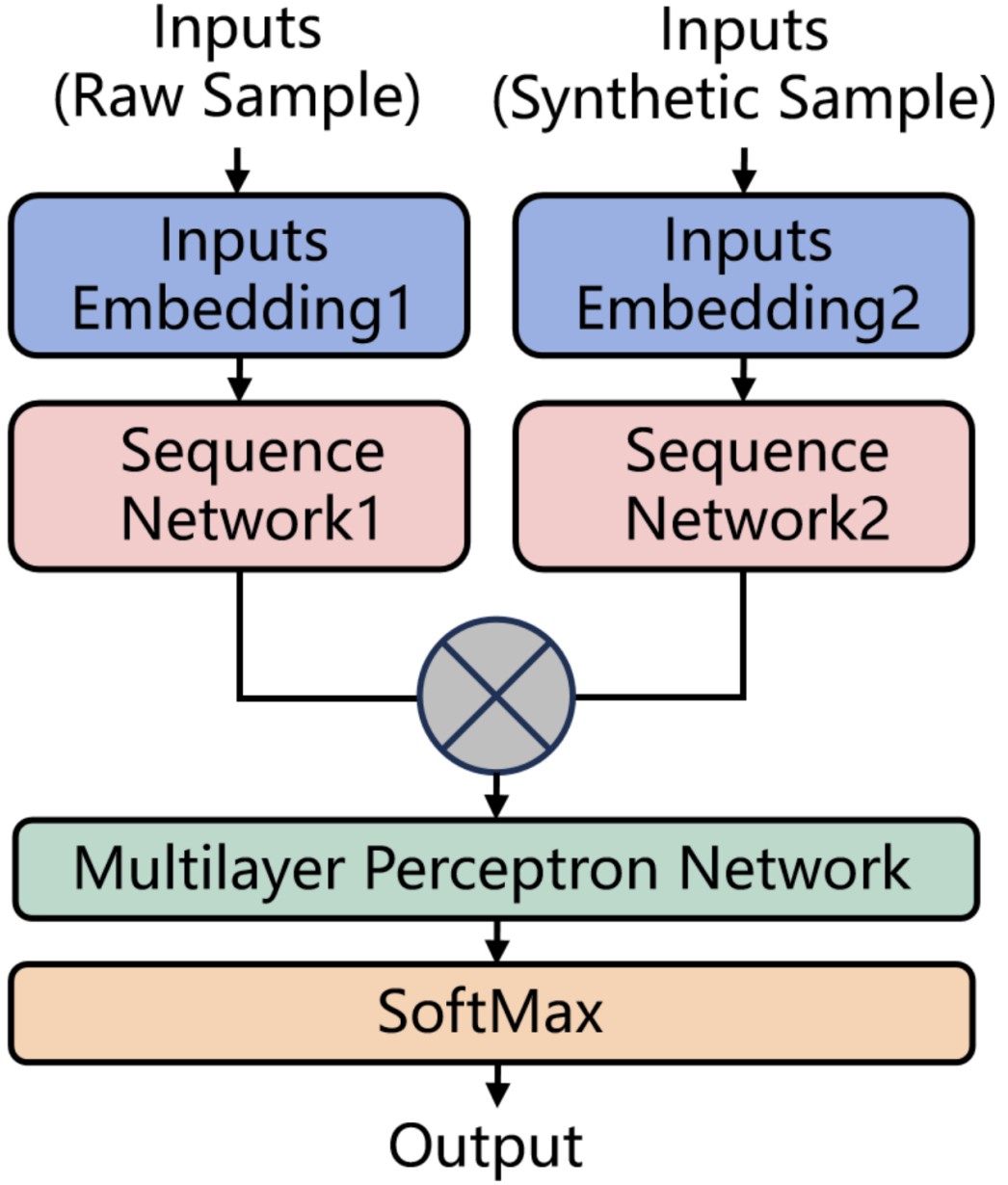

**Figure 5** The structure of the identification model.

## EXPERIMENTAL RESULTS

In this section, we conduct comparative experiments on three independent datasets in the first set of experiments, confirming that our method can indeed effectively enhance the detection performance of existing sequence models for heterogeneous IoT traffic samples. In the second set of experiments, we conduct a comparative evaluation of our method with the latest methods IoTDevID (*Kostas, Just & Lones, 2022*) and trace-clf (*Qu et al., 2023*) to

validate the advantages of our method in heterogeneous IoT traffic sample identification compared to similar algorithms.

## Datasets

We select three independent non-IP heterogeneous IoT traffic datasets for experimentation, and their descriptions are as follows:

- **CICIoT2022-ZigBee** (*Dadkhah et al., 2022*) dataset (http://205.174.165.80/IOTDataset/ CIC_IOT_Dataset2022/Dataset/) contains traffic samples generated by various ZigBee devices such as smart plugs, smart light bulbs, *etc.*, performing 14 different operations (*e.g.*, turning on lights, plug switches), with a total of 265 pcapng files. These files contain ZigBee packets transmitted through the ZEP protocol collected from the gateway. We develop a module in the heterogeneous sample extraction platform to read the ZigBee packets in order to generate sequence samples. Finally, the types of operations are used as sample labels, and the number of each label in the sample dataset is shown in Table 1.
- **CICIoT2022-ZWave** (*Dadkhah et al., 2022*) dataset (http://205.174.165.80/IOTDataset/ CIC_IOT_Dataset2022/Dataset/, https://github.com/narmeenshafqat1/ZLeaks/tree/ master/capture%20files) contains traffic samples generated by multiple Z-Wave devices during 13 different operations, totaling 145 zlf files. We first develop a Python script to call the Zniffer tool (this tool is included in Simplicity Studio, which is available at https://www.silabs.com/developers/simplicity-studio) to batch convert all zlf files to csv files, and then develop a corresponding module to read Z-Wave packets from the csv files to generate sequence samples. The number of its labels is shown in Table 2.
- **ZLeak-ZigBee** (*Shafqat et al., 2022*) dataset (https://github.com/narmeenshafqat1/ ZLeaks) contains traffic samples from seven different functional ZigBee devices, totaling 133 pcapng files. These samples are collected using sniffing tools directly from the wireless network, where ZigBee packets are transmitted through the IEEE802.15.4 protocol. We develop a module to extract ZigBee packets from the dataset to generate samples. The distribution of labels is provided in Table 3.

## Comparison with baselines

In order to validate that our method enhances the detection performance of existing sequence networks on heterogeneous IoT traffic samples, we design three groups for control experiments.

These networks include BiRNN, BiLSTM, BiLSTM-ATT, BiGRU, and CONV-1D. BiRNN (*Schuster & Paliwal, 1997*) learns features from both forward and backward sequences. BiLSTM (*Hochreiter & Schmidhuber, 1997*) incorporates memory cells to process sequences effectively. BiLSTM-ATT (*Lin et al., 2017*) enhances sequence representations using self-attention. BiGRU (*Chung et al., 2014*) leverages gate mechanisms to manage information flow and capture long-term dependencies. CONV-1D (*Zhang, Zhao & LeCun, 2015*) extracts local features for sequential data processing.

In the first group, we construct detection models using these five common sequence networks and feed them only with raw samples for classification. In the second group, we train the five models exclusively on the synthetic samples generated by the synthesis

**Table 1** The sample number of CICIoT2022-ZigBee dataset.

| Label | Manuf. | Device type | Events | Count |
|---|---|---|---|---|
| 0 | AeoTec | Button | Press | 5 |
| 1 | AeoTec | Motion Sensor | Motion&NoMotion | 10 |
| 2 | AeoTec | Multipurpose Sensor | Move | 5 |
| 3 | AeoTec | Multipurpose Sensor | Open&Close | 10 |
| 4 | AeoTec | Water Leak | Dry&Wet | 10 |
| 5 | Philips | Hue White | Increase&Decrease | 20 |
| 6 | Philips | Hue White | On&Off | 20 |
| 7 | Sengled | Smart Plug | LAN on & LAN off | 20 |
| 8 | Sengled | Smart Plug | PHY on & PHY off | 20 |
| 9 | SmartThings | Button | Press | 5 |
| 10 | SmartThings | Smart Bulb | Increase&Decrease | 50 |
| 11 | SmartThings | Smart Bulb | On&Off | 50 |
| 12 | Sonoff | Smart Plug | LAN on & LAN off | 20 |
| 13 | Sonoff | Smart Plug | PHY on & PHY off | 20 |

**Table 2** The sample number of CICIoT2022-ZWave dataset.

| Label | Manuf. | Device type | Events | Count |
|---|---|---|---|---|
| 0 | AeoTec | Door Window Sensor | Open&Close | 10 |
| 1 | AeoTec | Doorbell | LAN Chime | 5 |
| 2 | AeoTec | Doorbell | PHY Chime | 5 |
| 3 | AeoTec | Indoor Siren | LAN Chime | 5 |
| 4 | AeoTec | Nano Mote Quad | PHY Button | 5 |
| 5 | AeoTec | Smart Switch | LAN on & LAN off | 10 |
| 6 | AeoTec | Smart Switch | PHY on & PHY off | 10 |
| 7 | AeoTec | TemHumSensor | PHY Button | 5 |
| 8 | AeoTec | TriSensor | Motion & NoMotion | 10 |
| 9 | Fibaro | Door Window Sensor | Open&Close | 20 |
| 10 | Fibaro | Motion Sensor | Motion & NoMotion | 20 |
| 11 | Fibaro | Wall Plug | LAN on & LAN off | 20 |
| 12 | Fibaro | Wall Plug | PHY on & PHY off | 20 |

**Table 3** The sample number of ZLeak-ZigBee dataset.

| Label | Device type | Count |
|---|---|---|
| 0 | Audio:Ecolink Sound Sensor | 2 |
| 1 | bulb:BulbSengled Color,White Bulb *etc*. | 62 |
| 2 | Door:Visonic Door sensor *etc*. | 13 |
| 3 | Flood:Ecolink Water Sensor | 3 |
| 4 | Lock:Schlage Lock,Yale Door lock *etc*. | 10 |
| 5 | Motion:SMT Motion sensor *etc*. | 11 |
| 6 | plug:Centralite,Sonoff,SmartThings Outlet *etc*. | 54 |

algorithm. In the third group, we implement the sample identification framework proposed by our method, utilizing the same five sequence models as intermediate layers to build models and feed them with both raw samples and synthetic samples simultaneously.

The model structure used in the third group of experiments is shown in Fig. 5. Inputs Embedding Layer1 and Inputs Embedding Layer2 have identical structures, each with an output dimension of 32. For models constructed with BiRNN, BiLSTM, and BiGRU networks, their Sequence Network Layer1 consists of the corresponding networks, uses the tanh activation function, and has an output dimension of 64. For models constructed with BiLSTM-ATT, Sequence Network Layer1 consists of a BiLSTM layer with an output dimension of 64, a self-attention layer, and a flatten layer. For models constructed with the CONV-1D network, the structure consists of two Conv1D layers with 32 filters of size 3, a MaxPooling layer with a pool size of 5, two additional Conv1D layers with 32 filters of size 3, and a GlobalMaxPool layer as output. Sequence Network Layer2 has the same structure as Sequence Network Layer1. The outputs from both networks are concatenated and input into the MLP, which consists of an input layer, a hidden layer with 64 neurons, and an output layer with a number of neurons matching the number of labels. The activation functions used are ReLU for the hidden layers and SoftMax for the output layer. Additionally, for the first and second groups of experiments, Inputs Embedding Layer2 and Sequence Network Layer2 are not used.

For each dataset, five-fold cross-validation is performed to evaluate the models, with each model trained using 10 random seeds, and the average of the metrics is used as the result. The experiment uses accuracy as the metric to measure the model performance.

The experimental results are shown in Table 4. It can be seen from the results of the three comparative experiments, irrespective of the sequence model used, our method outperforms baseline sequence models trained with either raw samples or synthetic samples alone, achieving an average accuracy improvement of 29.7% compared to baseline models using only raw samples. Additionally, using synthetic samples is more effective than using raw samples directly. Comparative analysis of five types of sequence networks indicates that BiLSTM performed well on all three datasets.

## Comparison with the state-of-the-arts

To validate the advantages of our method over similar approaches in identifying heterogeneous IoT traffic samples, we conduct a comparative evaluation with the latest methods, IoTDevID (*Kostas, Just & Lones, 2022*) and trace-clf (*Qu et al., 2023*).

Trace-clf is an input-agnostic hierarchical deep learning framework for traffic fingerprinting that can hierarchically abstract comprehensive heterogeneous traffic features into homogeneous vectors seamlessly digestible by existing neural networks for further classification. The evaluation demonstrates that the method, with just one paradigm, not only supports heterogeneous traffic input but also achieves better or comparable performance compared to state-of-the-art methods (*Qu et al., 2023*). In this evaluation, we conduct experiments using the trace-clf method. To better handle ZigBee traffic, we optimized the method's pre-processing tool: (1) The optimized tool can directly read the ZigBee packet in the payload of ZEP, skipping the irrelevant UDP header; (2) Since

**Table 4  Performance comparison (accuracy in percentage): our method using raw and synthetic samples simultaneously versus baseline models using only raw or synthetic samples.**

| Dataset | Sequence model | Baseline model using only raw samples | Baseline model using only synthetic samples | Our method and improvement over baseline models with raw samples |
|---|---|---|---|---|
| CICIoT2022-ZigBee | BiLSTM-ATT | 50.2% | 77.7% | 83.8% (+33.6%) |
| CICIoT2022-ZigBee | CONV-1D | 44.5% | 75.9% | 84.2% (+39.7%) |
| CICIoT2022-ZigBee | BiGRU | 49.8% | 77.4% | 84.2% (+34.4%) |
| CICIoT2022-ZigBee | BiLSTM | 50.9% | 76.6% | 86.4% (+35.5%) |
| CICIoT2022-ZigBee | BiRNN | 54.0% | 74.7% | 84.2% (+30.2%) |
| CICIoT2022-ZWave | BiLSTM-ATT | 84.8% | 88.3% | 89.7% (+4.9%) |
| CICIoT2022-ZWave | CONV-1D | 85.5% | 82.1% | 87.6% (+2.1%) |
| CICIoT2022-ZWave | BiGRU | 86.9% | 88.3% | 89.7% (+2.8%) |
| CICIoT2022-ZWave | BiLSTM | 84.8% | 86.2% | 91.0% (+6.2%) |
| CICIoT2022-ZWave | BiRNN | 85.5% | 84.1% | 88.3% (+2.8%) |
| ZLeak-ZigBee | BiLSTM-ATT | 32.5% | 81.8% | 83.8% (+51.3%) |
| ZLeak-ZigBee | CONV-1D | 33.8% | 72.1% | 77.3% (+43.5%) |
| ZLeak-ZigBee | BiGRU | 29.9% | 83.8% | 85.7% (+55.8%) |
| ZLeak-ZigBee | BiLSTM | 29.2% | 82.5% | 86.4% (+57.2%) |
| ZLeak-ZigBee | BiRNN | 32.5% | 77.3% | 78.6% (+46.1%) |
| Average | – | 55.7% | 80.6% | 85.4% (+29.7%) |

the trace-clf method can directly read multidimensional inputs, the 4-tuple of packet information can be directly inputted into the model without coding into integers. (3) Following the trace-clf method's approach in processing the IoT SENTINEL dataset, we input the first two bytes of the packet payload into the sample.

IoTDevID is an ML-based method designed for the identification of IoT devices, utilizing generalizable packet-level features for device classification. The key contribution of this method is the use of a multi-stage feature selection process to determine a set of generalizable packet-level features. Additionally, the aggregation algorithm employed in this method leverages both the outcomes of the ML algorithm and the IP or MAC addresses of devices as input, thereby enhancing the effectiveness of device identification. Since (*Kostas, Just & Lones, 2023*) has evaluated the performance of IoTDevID on the CICIoT2022 dataset, we directly use the classification results of ZigBee devices in the confusion matrix provided by it to calculate the indicators in this evaluation.

Since IoTDevID can only handle ZigBee traffic transmitted *via* the ZEP protocol over UDP, this evaluation is limited to the CICIoT2022-ZigBee dataset, which contains nine ZigBee device types used as labels. On the evaluation metrics, since IoTDevID directly classifies a sample composed of 13 packets, while trace-clf and our method classify a sample composed of packets within fixed time intervals, in order to standardize the measurement unit of samples, the experiment uniformly uses the number of packets contained in the samples rather than the number of samples for performance evaluation. However, due to variations in preprocessing rules of traffic sample files among different methods, the

**Table 5 Performance comparison (in percentage): our method versus Trace-clf and IoTDevID.** The metrics are macro precision, macro recall, and macro F1-score.

| Metric | IoTDevID | Trace-clf | Our method and improvement over Trace-clf |
|---|---|---|---|
| Macro precision | 45.6% | 59.1% | 81.2% (+22.1%) |
| Macro recall | 41.0% | 60.5% | 82.0% (+21.5%) |
| Macro F1-score | 38.9% | 59.4% | 81.2% (+21.8%) |

number of packets contained in samples of different models is not completely equal, but the proportion of the same type of samples to the total samples is close, so the evaluation metrics of different models can still be roughly compared. Considering the significant differences in sample numbers between different devices when measured by the number of packets, we use the macro precision, macro recall, and macro F1-score as evaluation metrics.

In this evaluation, we use our method combined with the BiLSTM network to construct the model. The results of the comparative experiment are shown in Table 5. Our method achieves the best results in macro precision, macro recall, and macro F1-score, with improvements of 22.1%, 21.5%, and 21.8%, respectively compared to the trace-clf method. In contrast, IoTDevID performs comparatively lower.

## Discussion

The first set of experiments indicates that our method achieves an average accuracy improvement of 29.7% compared to baseline models using only raw samples. This is because our method is better adapted to the communication characteristics of IoT traffic. First, the Stop-and-Wait protocol used by the IoT devices causes duplicate packets to be retransmitted within the same sample, resulting in variations among similar network traffic samples. Meanwhile, due to the sleep strategy, IoT devices typically strive to remain in a sleep state to minimize battery consumption, which limits the number of traffic samples. This makes it challenging for the model to fit these specific sequence patterns of the Stop-and-Wait protocol from a limited number of communication samples. To address this issue, our method designs the synthesis algorithm to re-encode the repeated identical subsequences in the raw samples to generate new synthetic samples, thus mitigating the impact of the Stop-and-Wait protocol. The results from the experiments also reveal that the baseline model with synthetic samples achieves an average accuracy improvement of 24.9% compared to baseline models with raw samples. However, some devices have inherent duplicates in normal communication. The de-duplication process of the synthesis algorithm will also result in the loss of some information from the raw samples. Therefore, our method also simultaneously reads the raw samples as input, achieving the best results and an average accuracy improvement of 4.8% compared to baseline models with synthetic samples.

In the second set of experiments, we conduct a comparative evaluation of our method with the latest methods IoTDevID and trace-clf. Regarding the IoTDevID method, in the

case of heterogeneous IoT packet payload encryption, it mainly relies on the information in IP and UDP headers for classification. Since the sample packets are all encapsulated with ZEP and transmitted *via* UDP, this results in a high similarity in packet headers among different devices. This causes IoTDevID to struggle with similar encrypted traffic, resulting in relatively lower performance in the experiments. The weakness of the trace-clf method may lie in the large number of learnable parameters in its model. With a limited number of heterogeneous IoT samples, this method struggles to directly learn the communication characteristics of IoT traffic from the samples and encounters overfitting issues. However, trace-clf is able to extract more information from the sequence patterns of the packets, resulting in better performance compared to IoTDevID, with increases of 13.5%, 19.5%, and 20.5% in macro precision, macro recall, and macro F1-score, respectively. Since our method adapts to the communication characteristics of IoT traffic through the synthesis algorithm and mitigates the issue of overfitting with a smaller network structure, it achieves better performance compared to trace-clf. The experiments reveal that our method achieves improvements of 22.1%, 21.5%, and 21.8% in macro precision, macro recall, and macro F1-score, respectively, over the trace-clf method.

## LIMITATIONS

Firstly, by analyzing the experimental results, we observe significant confusion in the classification of IoT devices produced by the same company. This might be due to the fact that products from the same company could share the same code framework, leading to highly similar traffic samples. However, without access to plaintext, classifying traffic from IoT products developed with the same code framework is a challenging task. There is still room for improvement in our approach regarding this issue.

Secondly, in real-world attack and defense scenarios, there may be cases where IoT device keys become accessible. For example, manufacturers might leak fixed keys in the device firmware. However, when keys are available to decrypt plaintext payloads, our method lacks the appropriate framework to utilize this plaintext information.

Moreover, as discussed in the introduction, this method might also be used to probe users' personal privacy, potentially leading to additional security implications. Experiments indicate that current traffic encryption strategies do not fully address this issue, as packet sequence patterns may still reveal the device's behavior. Therefore, padding the end of packets with null characters to standardize their length and incorporating random heartbeat packets into the traffic may be effective methods for obfuscating packet sequence patterns. However, further experimental validation is needed.

## CONCLUSION

In this article, we propose a method for identifying IoT devices and events from non-IP heterogeneous IoT network traffic, which can be applied to scenarios such as network security and privacy leakage. In particular, we design a heterogeneous sample extraction platform that can shield the differences caused by the heterogeneous IoT protocol stack and support further expansion to other heterogeneous IoT protocols. Furthermore, we

propose a synthesis algorithm and the corresponding identification model based on the communication characteristics of IoT traffic. Our experiments are conducted on three independent heterogeneous IoT traffic datasets to validate that our method enhances the detection performance of the baselines on heterogeneous IoT traffic samples. The experimental results indicate that our method achieves an average accuracy improvement of 29.7% compared to baseline models using only raw samples. Additionally, we conduct a comparative evaluation of our method with the latest methods. Our method shows improvements of 22.1%, 21.5%, and 21.8% in macro precision, macro recall, and macro F1-score, respectively, over the trace-clf method.

A key contribution of this work is the proposal of a sample identification framework, which includes the synthesis algorithm and the identification model. According to the experimental discussion, the synthesis algorithm can re-encode repeated identical subsequences in the raw samples to generate new synthetic samples, thereby mitigating the effects of retransmitted duplicate packets. Furthermore, the identification model employs two independent sequence networks that simultaneously handle both raw and synthetic samples, resulting in optimal performance in the experiments.

In future work, we plan to employ multimodal technology to enable the model to simultaneously analyze both the sequence patterns of traffic and the textual content of packets, thereby improving the performance of the method and expanding its applicability.

### Funding
This work is funded by the Special Fund for Scientific Research of Fujian Provincial Department of Finance in 2022, the Fujian Provincial Department of Education Young Teachers Education Research Project (JAT210354). The funders had no role in study design, data collection and analysis, decision to publish, or preparation of the manuscript.

### Grant Disclosures
The following grant information was disclosed by the authors:
Special Fund for Scientific Research of Fujian Provincial Department of Finance in 2022.
Fujian Provincial Department of Education Young Teachers Education Research Project: JAT210354.

### Competing Interests
The authors declare there are no competing interests.

### Author Contributions
- Yi Chen conceived and designed the experiments, performed the computation work, prepared figures and/or tables, authored or reviewed drafts of the article, and approved the final draft.
- Junxu Lai performed the experiments, performed the computation work, prepared figures and/or tables, and approved the final draft.

- Zhu Lin analyzed the data, prepared figures and/or tables, and approved the final draft.
- Meijing Zhang conceived and designed the experiments, authored or reviewed drafts of the article, and approved the final draft.
- Wenxi Liu conceived and designed the experiments, authored or reviewed drafts of the article, and approved the final draft.

## Data Availability

The ZLeak-ZigBee dataset is available at GitHub: https://github.com/narmeenshafqat1/ZLeaks/tree/master/capture files

The CIC IoT dataset 2022 (ZigBee and ZWave) from the Canadian Institute for Cybersecurity are available at: https://www.unb.ca/cic/datasets/iotdataset-2022.html.

## Supplemental Information

Supplemental information for this article can be found online at http://dx.doi.org/10.7717/peerj-cs.2363#supplemental-information.

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
