# Peer review of "Identify devices and events from non-IP heterogeneous IoT network traffic"

_PeerJ Computer Science, doi:10.7717/peerj-cs.2363_

## Round 0.1 · original submission · Minor Revisions

After the first round of review, this paper is found interesting and can be considered to publish in this journal based on the minor revision based on the reviewers' comments. Authors need to address the following comments.

1. in the "Abstract", authors should provide the quantitative performance indicators of the proposed work.
2. In the "Conclusion", authors reflect on what have been achieved based on the performance of the proposed work.
3. I can see some references were asked by the reviewer to cite in the revision. Please read and carefully check if there are not relevant, please don't add or cite in your paper.

Reviewer 1 ·

Basic reporting

Identify Devices and Events from Non-IP Heterogeneous IoT Network Traffic is addressed in this work.
• The caption of Figs. is too large and may be precise and concise.
• Enhancing the paper by thoroughly addressing grammatical and spelling errors will improve its clarity and overall quality.
• Simplify complex sentences and reduce technical jargon to improve readability and make the paper more accessible to a broader audience.
• Clearly articulate how the proposed mechanism directly addresses the identified problems. This will help in reinforcing the connection between the identified problems and the proposed solutions.

Experimental design

• Result section is not adequate to justify the work. Add some more results and Fig. 7 may explain in details about results values.

Validity of the findings

• Also elaborate security implications and overcome with your proposed approach.
• Also define limitations and novelty of your work.

Additional comments

Limitations can be modified in Conclusion section and future work may be clearly defined.

Cite this review as

Reviewer 2 ·

Basic reporting

The Introduction referred to some references related to the security of IoT networks, which are out of context to this article.

It is crucial to define the Variables used in the algorithms, such as "N" in algorithm 2,

All Graphs showing the comparison, the units used to refer to the performance need to be Explained

Experimental design

In Figure 2, it is not clear how a file is recognized while data transmission is done using packets using different protocols. This needs to be addressed properly.

Validity of the findings

Clarity is required concerning identifying a device using a "label" while the devices are pre-known as claimed to have copied the sample data to different device directories, each relating to a different device/


It is not clear why raw samples are to be processed while synthetic samples are generated by processing the raw samples. Synthetic samples are, as such, preprocessed samples.

It is important to provide details of the design of each networking method used in the research, such as the number of layers used and the methods used in each layer. The SoftMax algorithm is indicated as being used to convert the processed data to produce the output considering each of the models. It needs Justification here.

Additional comments

The paper is well written and has expressed an excellent novelty of considering low paper non-IP-Devices for effecting communication between heterogeneous IoT devices

Cite this review as

Reviewer 3 ·

Basic reporting

Please read the general comments

Experimental design

Please read the general comments

Validity of the findings

Please read the general comments

Additional comments

The paper proposes an approach to identify IoT devices and events in non-IP heterogeneous
networks by extracting and processing raw and synthetic traffic samples from protocols like
ZigBee and Z-Wave. The idea is interesting, and the paper is well-written and presented. However,
some improvements are needed before acceptance:
1. It is recommended that the authors diversify their result presentation by incorporating
tables that show numerical values and the percentage of improvements for better
readability.
2. In the abstract, please highlight the level of improvements obtained by your method by
including percentage values. This should also be reflected in the conclusion.
3. I suggest that captions for all figures be brief and simple, moving detailed descriptions to
the main text without repetition.
4. The authors mention the datasets used in references no. 8 and 14. However, they must
include the direct link to the dataset in the main text of the paper.
5. Authors should also declare any other resources that are accessible only to users and make
them accessible to readers.

Annotated reviews are not available for download in order to protect the identity of reviewers who chose to remain anonymous.
Cite this review as

---

## Round 0.2 · accepted · Accept

Authors have addressed all the comments to satisfy all the reviewers. This paper is now recommended to publish in its current form.

Reviewer 2 ·

Basic reporting

Every thing is properly addresses

Experimental design

Experimental design is fine

Validity of the findings

The Findings are Valid

Additional comments

Excellent work done

Cite this review as

Reviewer 3 ·

Basic reporting

The authors have adequately responded to all my comments

Experimental design

The authors have adequately responded to all my comments

Validity of the findings

The authors have adequately responded to all my comments

Additional comments

The authors have adequately responded to all my comments

Cite this review as